# Characterization of Porcine Ocular Surface Epithelial Microenvironment

**DOI:** 10.3390/ijms24087543

**Published:** 2023-04-19

**Authors:** Naresh Polisetti, Gottfried Martin, Heidi R. Cristina Schmitz, Ursula Schlötzer-Schrehardt, Günther Schlunck, Thomas Reinhard

**Affiliations:** 1Eye Center, Medical Center, Faculty of Medicine, University of Freiburg, Killianstrasse 5, 79106 Freiburg, Germany; 2CEMT-Freiburg, Experimental Surgery, Hospital—Medical Center, Faculty of Medicine, University of Freiburg, Breisacher Str. 66, 79106 Freiburg, Germany; 3Department of Ophthalmology, University Hospital Erlangen, Friedrich-Alexander-University of Erlangen-Nürnberg, Schwabachanlage 6, 91054 Erlangen, Germany

**Keywords:** limbal stem cells, limbal niche cells, melanocytes, limbal epithelial progenitor cells, porcine ocular surface, extracellular matrix, cornea, conjunctiva

## Abstract

The porcine ocular surface is used as a model of the human ocular surface; however, a detailed characterization of the porcine ocular surface has not been documented. This is due, in part, to the scarcity of antibodies produced specifically against the porcine ocular surface cell types or structures. We performed a histological and immunohistochemical investigation on frozen and formalin-fixed, paraffin-embedded ocular surface tissue from domestic pigs using a panel of 41 different antibodies related to epithelial progenitor/differentiation phenotypes, extracellular matrix and associated molecules, and various niche cell types. Our observations suggested that the Bowman’s layer is not evident in the cornea; the deep invaginations of the limbal epithelium in the limbal zone are analogous to the limbal interpalisade crypts of human limbal tissue; and the presence of goblet cells in the bulbar conjunctiva. Immunohistochemistry analysis revealed that the epithelial progenitor markers cytokeratin (CK)15, CK14, p63α, and P-cadherin were expressed in both the limbal and conjunctival basal epithelium, whereas the basal cells of the limbal and conjunctival epithelium did not stain for CK3, CK12, E-cadherin, and CK13. Antibodies detecting marker proteins related to the extracellular matrix (collagen IV, Tenascin-C), cell–matrix adhesion (β-dystroglycan, integrin α3 and α6), mesenchymal cells (vimentin, CD90, CD44), neurons (neurofilament), immune cells (HLA-ABC; HLA-DR, CD1, CD4, CD14), vasculature (von Willebrand factor), and melanocytes (SRY-homeobox-10, human melanoma black-45, Tyrosinase) on the normal human ocular surface demonstrated similar immunoreactivity on the normal porcine ocular surface. Only a few antibodies (directed against N-cadherin, fibronectin, agrin, laminin α3 and α5, melan-A) appeared unreactive on porcine tissues. Our findings characterize the main immunohistochemical properties of the porcine ocular surface and provide a morphological and immunohistochemical basis useful to research using porcine models. Furthermore, the analyzed porcine ocular structures are similar to those of humans, confirming the potential usefulness of pig eyes to study ocular surface physiology and pathophysiology.

## 1. Introduction

The ocular surface is composed of three phenotypically and functionally distinct structures: the cornea, the limbus, and the conjunctiva. The cornea is a unique, transparent, avascular structure that is covered by the corneal epithelium, which provides a tight protective barrier. The conjunctiva covers the anterior sclera, which encircles the cornea, and extends to cover the inner surface of the eyelids. Conjunctival goblet cells secrete mucins as an essential component of the tear film, the healthy conjunctiva allows for eye movements, and its associated lymphoid tissue provides an important pathogen defense system [1]. The limbus is a narrow circular band of tissue that separates the cornea and the conjunctiva and contains limbal epithelial stem/progenitor cells (LEPC), which are responsible for homeostasis of the corneal epithelium [2,3,4,5]. Unlike LEPC, conjunctival epithelial stem/progenitor cells (CjEPC) are dispersed throughout the conjunctival epithelium, with the greatest cell numbers in the medial bulbar and inferior forniceal regions [6]. The corneal, limbal, and conjunctival epithelia, the neural network that supports them, as well as other components such as immune cells, the extracellular matrix (ECM), and neighboring cells interact to form the ocular surface microenvironment, which is pivotal for a healthy ocular surface.

Severe ocular surface disease can lead to limbal stem cell deficiency (LSCD), a condition characterized by persistent corneal epithelial breakdown, superficial corneal vascularization, chronic discomfort, and impaired vision caused by the migration of conjunctival epithelial cells and blood vessels onto the corneal surface [7]. Depending on the pathological conditions, many treatments have been employed to treat LSCD, including the transplantation of ex-vivo-expanded LEPC [4,8]. Regardless of the reasons for LSCD or the transplantation procedures used, LEPC treatment is clinically successful in 50–70% of reported patients [9]. The long-term success rate is constrained by the quality of LEPC in the graft and the short survival times of transplanted cells. Therefore, the ability of transplanted cells to maintain an undifferentiated limbal phenotype and to respond to injury remains unclear. Although there are many animal models (rabbit, rat, mouse) of LSCD that have been examined [10,11,12], several reports indicate that LEPC are not confined to the limbus in these animals [13], which contrasts with the spatial restriction of LEPC in humans. In this regard, the porcine ocular surface has been shown to be quite comparable to the human ocular surface and interpalisade crypt structures have been revealed in porcine tissue [14,15,16]. Furthermore, the porcine ocular surface has been used as in vitro organ culture model to investigate the fate of transplanted human limbal epithelia as well as the limbal response to injury [14,16]. However, no reports on porcine LSCD models have been published thus far, owing in part to a lack of detailed characterization of the ocular surface epithelial microenvironment. Furthermore, a thorough characterization of the porcine ocular surface is warranted as pig eyes are used in research on the corneal penetration of nanoparticles and liposomes as well as on corneal wound healing.

Few studies have documented a partial characterization of the porcine ocular surface microenvironment, with more focus on the epithelium [14,15,17]. There has not been a detailed characterization of the ocular surface epithelium including the ECM and neighboring cells. This is due, in part, to the scarcity of antibodies produced specifically against porcine ocular surface cell types or structures. Because it is expected that an increasing number of experimental studies will use porcine models to study human-ocular-surface-related diseases in the future, there is a need for a better understanding of the histological and immunohistochemical properties of the normal porcine ocular surface epithelial microenvironment. Therefore, the goal of this study is to provide a thorough description of the pig ocular surface confined to the cornea, limbus, and bulbar conjunctiva, with a particular focus on the limbal stem cell niche. A panel of antibodies, most of them specific to the human ocular surface, were tested in order to investigate which of them would also provide reliable labeling patterns in porcine ocular surface tissues and could therefore be used for histopathological investigations in porcine LSCD and other ocular surface disease models.

## 2. Results and Discussion

### 2.1. Architecture of Porcine Ocular Surface

In recent years the use of porcine eyes has increased because of the similarity of porcine and human eye morphology [14,15,18], which makes the pig a very useful model to study ocular surface diseases such as LSCD. In this study, the morphological and anatomical characteristics of the porcine ocular surface were investigated by means of light and electron microscopy. A dashed line was used to represent the epithelial basement membrane (BM) if it was not delineated by stained marker proteins. Hematoxylin and eosin (H&E) staining of porcine ocular surface tissue sections revealed that the corneal epithelium consists of 6–8 layers of stratified epithelium and Bowman’s layer is not evident; deep invaginations of the limbal epithelium (6–12 layers) analogous to the limbal interpalisade crypts of human limbal tissue can be observed in the limbal zone; the bulbar conjunctiva is composed of a 3–4-layered stratified epithelium (Figure 1A). The conjunctival and limbal stroma are organized differently to the corneal stroma, with loose connective tissue, a large number of fibroblasts, and a significant presence of vessels (arrows, Figure 1A). These findings are consistent with previous studies [14,15,18]. Alcian blue (AB) staining of the ocular surface showed the presence of glycosaminoglycans (blue) throughout the cornea and a rather weak staining in the limbal and conjunctival stroma (Figure 1A). As observed in an earlier publication [15], the bulbar conjunctiva has very few goblet cells (arrows, Figure 1A) and their number increases towards the conjunctival fornix, whereas the tarsal conjunctiva contains a higher number of goblet cells stained with periodic acid-Schiff (PAS) (pink) and AB (blue, arrows, Appendix A). PAS staining highlighted the BM of the porcine corneal epithelium (arrows) and was less pronounced in the epithelial BM of the limbus and conjunctiva (Figure 1A). The differences in epithelium thickness could be attributed to different donor tissues used in the study.

The ultrastructure of the porcine ocular surface was evaluated by transmission electron microscopy (TEM). TEM analysis showed the corneal epithelium consists of 6–8 multilayered epithelium with the columnar basal epithelium and no indication of Bowman’s layer (Figure 1B(i)). In contrast to human limbal epithelial progenitor cells (cuboidal) [5], the basal epithelial cells of the porcine limbus were slender (Figure 1B(ii)), elongated cells forming extensive interdigitations of their lateral cell membranes and irregular basal cell surfaces extending into the limbal stroma (Figure 1B(iv)). The limbal epithelium intermingled with infiltrating immune cells (arrows, Appendix A) and LEPC were in close contact with subepithelial stromal immune cells (arrows, Figure 1B(v)) and mesenchymal stromal cells (Figure 1B(vi)). Goblet cells (arrows, Figure 1B(iii)) intercalated within the stratified epithelium of the bulbar conjunctiva.

### 2.2. Immunohistochemical Characterization of the Porcine Ocular Surface

Tissues for research can be collected from laboratory animals, with the most commonly utilized species in ophthalmology-related investigations being rats, mice, and rabbits. As a result, the vast majority of commercially available antibodies are raised against these species, while just a few are raised/tested against porcine antigens [14,15]. In the present study, we aimed to study a panel of 41 different antibodies to recognize the structure, differentiation, and proliferation-associated antigens of the porcine ocular surface. For sake of clarity, the markers used in this study were categorized into three different families, i.e., epithelial progenitor/differentiation markers, ECM and associated molecules, and niche-cell-related markers. All antibodies were tested in triplicate using domestic porcine samples (*n* = 5), wild pigs (*n* = 3), and Aachen minipigs (*n* = 3), with representative images provided.

#### 2.2.1. Epithelial Progenitor/Differentiation Markers

Immunohistochemical staining of paraffin sections showed the expression of epithelial keratins (pan-CK) in all epithelial layers of the cornea, limbus, and conjunctiva (red, Figure 2). Paired box 6 (Pax6), a master protein for eye development, is essential for maintaining proper differentiation and strong intercellular adhesion of the ocular surface epithelium [19,20]. The present study confirmed nuclear Pax6 expression in all layers of the corneal epithelium and in nearly all epithelial cell layers of the limbus and conjunctiva except for some basal cells (white arrows, Figure 2). These Pax6 negative cells might be either Langerhans cells or melanocytes [21,22]. Occasionally, Pax6 negative cells were also observed in superficial epithelial layers of the cornea, limbus, and conjunctiva (black arrows, Figure 2), suggesting that Pax6 may play a role in the shedding of the ocular surface epithelium. No Pax6 expression was detected in the stroma of the cornea, limbus, and conjunctiva (Figure 2), similar to observations made in human corneal and scleral surfaces [22]. Cytokeratin (CK) 12, a cornea-specific keratin, was expressed in all layers of the corneal epithelium, but only in suprabasal cell layers of the limbal epithelium (arrows, Figure 2), in line with a previous study [14]. As expected, CK12 expression was not detected in the conjunctival epithelium (Figure 2). Conjunctival-epithelium-specific CK13 was present in the suprabasal cells of the limbus and conjunctiva but completely absent in the cornea (Figure 2). Double immunostaining of corneoconjunctival frozen sections revealed co-localization of CK12 (red) and CK13 (green) in the suprabasal layers of the limbal epithelium, with the expression of CK13 increasing towards the conjunctiva (Appendix A), thus correlating with the equivalent human phenotype [23,24,25]. Expression of the epithelial progenitor marker CK14 was reported in the basal layers of the human cornea, limbus, and conjunctiva [26]. In the present study, CK14 was present in the basal layers of the limbal and conjunctival epithelia (arrows) but was negative in the corneal epithelium (Figure 2). The lack of corneal CK14 expression might be due to species variability. Interestingly, another epithelial progenitor marker, CK17/19, was present in the basal layers of the limbal epithelium (arrows, Figure 2) similar to the human corneo-limbus [5], but it was also present in all layers of the conjunctival epithelia (Figure 2). p63α, a known putative LEPC marker for human cells [27,28,29], was detected in the basal layers of the corneal, limbal, and conjunctival epithelia, but the expression was stronger in the limbus and conjunctiva (Figure 3A). A few cells in the limbal and conjunctival basal layers were negative for p63α (arrows, Figure 3A) and could represent melanocytes. The proliferation marker Ki-67 was present in the basal cells of the cornea, limbus, and conjunctiva. However, the limbus and conjunctiva contained more Ki-67^+^ cells than the cornea (arrows, Figure 3). These data suggest that epithelial progenitor cells are more frequently detected in the limbus and are responsible for the homeostasis of the corneal epithelium.

Antibodies which did not show reactivity on paraffin sections or which showed non-specific labeling on paraffin sections using alkaline phosphatase detection were examined on frozen sections using fluorescent labeling. The superficial layers of epithelial cells in both the limbus and conjunctiva expressed the differentiation marker E(epithelial)-cadherin (red) with rather weak/no expression in the basal cells (arrows), whereas all layers of the corneal epithelium expressed E-cadherin (Figure 3B). In contrast, the basal layers of the limbal and conjunctival epithelia expressed the epithelial progenitor marker CK15 (arrows, red) but not in the corneal epithelia (Figure 3B). Similarly, the basal layers of the limbal and conjunctival epithelia showed uniform immunoreactivity for P(placental)-cadherin (arrows, green, Figure 3B), which has been used for the efficient isolation of LEPC in humans [30], implying that P-cadherin could be used as a selective marker for the isolation of porcine LEPC. The progenitor marker expression patterns (CK15, P-cadherin, CK14, CK17/19) suggest that LEPC are distributed uniformly across the porcine limbus, in contrast to humans, where LEPC clusters have been described [31]. Interestingly, anti-N(neural)-cadherin antibody, which readily stains human LEPC in the human limbus [30], proved unreactive to the porcine limbal/conjunctival basal epithelium [32]. The expression of CK3/76 (green, Figure 3B), a corneal epithelial marker, was comparable to that of CK12, corresponding with an identical human phenotype [5]. These data suggest that almost all antibodies raised against human epithelial progenitor/differentiation markers were reactive on porcine ocular surface tissue and can be used to study ocular diseases in a porcine model. Furthermore, the porcine epithelial phenotype is remarkably similar to that of humans, more so than any other animal previously studied [14]. Table 1 contains a list of epithelial markers and their expression patterns on porcine ocular surface epithelial cells.

#### 2.2.2. ECM and Its Associated Markers

The stem/progenitor cells reside in a specialized microenvironment (niche) that regulates the function of stem/progenitor cells through interactions with the ECM (cell–matrix interactions) and neighboring cells (cell–cell interactions) either directly or through the use of soluble growth and signaling factors [33]. Previous studies of human limbal tissue demonstrated the presence of a limbus-specific ECM and BM containing collagen (Col)IV, Tenascin(TN)-C, fibronectin, and laminin (LN) isoforms [31,34,35,36]. We investigated the presence of these molecules on the porcine ocular surface tissue. The anti-human ColIV antibody strongly stained limbal BM (arrows) and elicited a very weak staining in corneal and conjunctival epithelial BM (Figure 4A), suggesting a pronounced limbal enhancement of ColIV in the pig. The BM of vessels in the limbal and conjunctival stroma also stained for ColIV (arrowheads, Figure 4). Tenascin-C, a subepithelial matrix component, has been found in the corneo-limbal transition zone and demonstrated significantly increased deposition underneath the limbal epithelium in aniridia-associated keratopathy [31,37]. In the current study, TN-C staining was observed in the posterior stroma (further away from the epithelium) of the limbus and conjunctiva (red) but not in the corneal stroma (Figure 4B), as well as in the anterior stroma of the limbus and conjunctiva [38]. The other ECM-related antibodies that readily stain the human ocular surface [31], such as fibronectin, agrin, LN-α3, and LN-α5, were unreactive to the porcine ECM [39]. Dystroglycan, a cell adhesion molecule for BM anchorage, forms a critical link between the ECM (mainly laminin α2 chain and agrin) and the cytoskeleton of muscle cells but is also present in non-muscular tissues including the brain, skin, and hematopoietic stem cells [40]. In the human limbal stem cell niche, dystroglycan was found to localize to the basal cell membrane of LEPC and occasionally also to subepithelial stromal niche cells [31]. In the present study, the antibody against β-dystroglycan was detected in the basal cell membranes of the limbus and conjunctiva, whereas faint membrane staining was present in the superficial cells of the limbus and conjunctiva as well as the corneal epithelium (Figure 4C). Interestingly, β-dystroglycan staining was prominent in the limbal stroma but weak in the conjunctival stroma and absent in the corneal stroma (Figure 4C). Integrins, ECM-binding cell surface receptors, play a major role in various stem cell niches including the limbal one by guiding stem cell niche architecture, regulating stem cell proliferation and self-renewal to maintain stem cells in the niche, and, finally, controlling the orientation of dividing stem cells [41,42]. Integrin α3 staining was present in the basal aspect of the corneal, limbal, and conjunctival epithelia, with the limbus and conjunctiva showing a strong expression (green, Figure 4D). The hemidesmosomal integrin α6 occurred in all ocular surface epithelia, with a continuous strong basal membrane staining and a rather weak signal in lateral cell membranes (arrows); it was also present in the vasculature of the limbal and conjunctival stroma (arrowheads, Figure 4D). These findings are comparable to those reported for human ocular surface tissue [5,31], but for detailed characterization of the ECM of the porcine ocular surface, the unreactive antibodies need to be replaced by antibodies raised against porcine proteins.

#### 2.2.3. Limbal Niche Cell Markers

In addition to a linkage to ECM components, the LEPC maintain close contact with various supporting cell types including melanocytes (LM), mesenchymal stromal cells (LMSC), vascular cells, immune cells, and nerves across a fenestrated epithelial BM [31,41,43,44,45]. These non-epithelial limbal niche cells nourish, protect, and regulate the quiescence, self-renewal, and fate decisions of limbal epithelial progenitor cells, LEPC [43,46]. It has been reported that porcine LMSC support the growth potential of LEPC and exhibit cross-reactivity with human MSC markers such as CD90, CD105, CD146, and human leukocyte antigen (HLA)-ABC [47]. The porcine cross-reactivity of antibodies raised against the human stromal cell antigens vimentin, HLA-ABC, CD44, and CD90 was investigated. Vimentin was found in the stromal cells of the porcine cornea, limbus, and conjunctiva (red, Figure 5A). We also found vimentin^+^ cells in the epithelial layers of both the limbus and conjunctiva (arrowheads, Figure 5A), suggesting the presence of melanocytes [45]. Expression of CD44, a hyaluronate receptor involved in the cell–cell and cell–matrix interactions, was reported in the basal cells of the ocular epithelia, stroma, and vasculature [48]. In the current study, CD44 was found on the plasma membranes of basal epithelial cells and stromal cells of the cornea (red), with predominant expression in the limbus and conjunctiva (Figure 5B). Further research into the low expression of CD44 in the corneal stroma is required. The stromal cells of the limbus and conjunctiva expressed the mesenchymal marker CD90 (red, Figure 5B), which has been used to isolate LMSC [30,49], suggesting that anti-human CD90 could be used to isolate porcine LMSC. An antibody raised against neurofilament (NF) was used to detect porcine nerves, nerve terminals, and axons, and its expression was observed in the stroma of all three surface regions (arrowheads, Figure 5B). Interestingly, the limbal BM was stained for NF along with nerves in the stroma, whereas the conjunctival BM showed only faint staining and the corneal BM showed no staining (arrowheads, Figure 5B). However, the immunostaining of paraffin sections using alkaline phosphatase labeling did not demonstrate an NF signal on the BM of the ocular surface (Appendix A). This raises the question as to whether NF staining at the limbal epithelial BM may be an artifact or cross-reactive binding, which warrants further studies. The anti-human HLA-class I antibody stained all cells of the ocular surface epithelium and stroma (detected swine leukocyte class I, Figure 5B), whereas swine leukocyte class II^+^ cells (detected by HLA-class II antibody) were present in the epithelial layers and stroma of the limbus and conjunctiva (arrows, Figure 6A).

The antibody to S100 protein, which identifies human Langerhans cells (LC), melanocytes, and Schwann cells, did not show reactivity on the porcine epidermis [50]. In the present study, S100 was present in a few cells of the limbus and the conjunctival epithelial basal layer, suggesting melanocytes (arrowheads, Figure 6A). The S100^+^ cells were also found in the stroma of all three regions (arrows, Figure 6A). The vasculature of the porcine ocular limbus and conjunctiva was labeled with anti-von Willebrand factor (VWF) (arrows, Figure 6A). CD4, a cell surface receptor found on the surface of the immune cells such as T helper cells, monocytes, macrophages, and dendritic cells, was detected in a few cells of the limbus and conjunctiva stroma (arrows, Figure 6A). LC are specialized antigen-presenting cells within the multilayered epithelia and play a critical role in regulating cutaneous immune responses [51]. These cells were identified using an antibody specific for CD1a antigen and found within the multilayered epithelium of the limbus and conjunctiva (arrows) but not in the cornea (Figure 6B). CD14, a lipopolysaccharide receptor complex protein, has been found on the surfaces of monocytes, macrophages, and polymorphonuclear leukocytes, as well as human corneal epithelial, stromal, and endothelial cells [52]. In contrast to humans, the current study on the porcine ocular surface found it was expressed in a few cells of the limbal and conjunctival stroma (arrows, Figure 6B) but not in the ocular surface epithelia, suggesting that expression varies between species.

Melanocytes, melanin-producing cells, are specialized cells of neural crest origin residing within the basal layer of the limbal and conjunctival epithelium of the ocular surface. Melanocytes can be identified in the human limbus with histochemical stains for melanin (Fontana-Masson staining), and more reliably with antibodies recognizing melanocyte-specific antigens such as melanoma antigen recognized by T cells 1 (MART1)/Melan-A; SRY-homeobox-10 (SOX-10), the nuclear transcription factor involved in melanocyte maturation; the premelanosome-associated glycoprotein 100 (gp100/human melanoma black-45(HMB-45)); and tyrosinase (TYR) and tyrosinase-related protein 1 (TRP1), the enzymes involved in melanin biosynthesis. Fontana-Masson staining revealed the presence of melanin in ocular surface tissue in only one of the five porcine samples in our investigation (Figure 7A). The anti-LM antibodies MART1/Melan-A and TRP1, which readily stain melanocytes in the human limbus, proved unreactive to porcine limbal/conjunctival melanocytes [53], similar to prior reports on porcine skin [50]. In line with the obtained findings by Fontana-Masson staining, the additional LM markers S0X10, HMB-45, and TYR stained melanocytes in the basal layers of the limbal and conjunctival epithelia (arrows, Figure 7B) in only one of the samples. We also tested the ocular surface of Aachen minipigs for the existence of melanocytes, which did not reveal any melanin (Figure 7A), nor were the above mentioned melanocyte markers detected [54]. A report on porcine skin (Seghers hybrid white) also indicated that melanocyte markers were unreactive on porcine skin [50] but reported vimentin^+^ cells in the basal layer of the porcine epidermis. We also analyzed the wild pig porcine ocular surface for clarification, which revealed melanin pigmentation (arrows, Figure 7A) and antibody reactivity in all samples (Appendix A). This suggests that the porcine ocular surface may contains melanocytes (in the limbal and conjunctival region); however, this is at a density that differs significantly according to the limbal area examined, and possibly also with animal strain.

It can also be speculated that at least some of the melanocytes could be resting, non-functional melanocytes, not expressing more differentiated antigens recognized by the anti-human melanocyte antibodies tested. TEM analysis revealed that the presence of basal limbal epithelial cells (LEPC) in association with LM (Figure 8) at the corneo-limbal transition zone (Figure 8(i) (dotted square bracket indicates the melanocyte) and Figure 8(ii)), limbal palisade zone (Figure 8(iii) (arrow indicates melanocyte) and Figure 8(v)), and the presence of melanosomes within the LEPC (arrows, Figure 8(iv)) in only one of the four porcine samples (from slaughterhouse) studied. In other samples, atypical cells in the basal limbal epithelium were occasionally observed (arrow, Figure 8(vi)), without any melanosomes but also without any clear relationship to immune cells, implying that these were atypical melanocytes. Nonetheless, further research is needed to investigate these cells. In addition, genetic analysis of wild and mini pigs would aid in a better understanding of the existence of melanocytes. In total, 41 different antibodies were tested, and 33 of them exhibited immunoreactivity. However, our investigation is confined to the immunohistochemical characterization of the porcine ocular surface. Additional studies on the transcriptomic/proteomic profile of the porcine ocular surface will help in further understanding the physiology of the porcine ocular surface.

In conclusion, we provide a detailed characterization of the porcine ocular surface epithelium as well as the surrounding microenvironment including the ECM and neighboring cells using human-specific antibodies. Moreover, we demonstrate a strong similarity of the analyzed porcine ocular structures to those of humans, strongly suggesting the potential usefulness of pig eyes to study ocular surface physiology and pathophysiology.

## 3. Materials and Methods

### 3.1. Porcine Eyes

Porcine eye balls (*n* = 10) of domestic pigs (sus scrofa domesticus) were obtained from local slaughterhouse (Freiburg, Germany). We also obtained eye balls of wild pigs (*n* = 3) donated by a licensed hunter (Freiburg, Germany) and eye balls of Aachen minipigs (*n* = 3) from the Center for Experimental Models and Transgenic mice, University of Freiburg. Porcine eyes were obtained within 12 h postmortem and transported to the laboratory on ice.

### 3.2. Tissue Preparation and Histology

Corneal–limbal–conjunctival (bulbar) tissue was excised from the eyeball using scalpel and scissors and processed for histology. Tissue was either fixed in 4% paraformaldehyde (30 min) and embedded in paraffin or embedded and frozen in an optimal cutting temperature (OCT) medium. Sections of five micrometer thickness were cut and stained as described previously [55]. Briefly, sections were stained with hematoxylin (Haematoxylin Gill III, Surgipath, Leica, Germany) for 2 min and 1% eosin Y (Surgipath, Leica, Germany) (H&E) for 1 min to observe the gross tissue architecture. To visualize the proteoglycan content and goblet cells, AB and PAS staining were performed. Sections were stained with 1% AB (Morphisto, Offenbach am Main, Germany) for 30 min together with a counterstain of nuclear fast red (Morphisto, Offenbach am Main, Germany). PAS staining was performed using 1% periodic acid (Honeywell Fluka, Charlotte, NC, USA) for 10 min, and Schiff reagent (Roth, Karlsruhe, Germany) for 90 s. For melanin detection, Fontana-Masson staining was performed according to the manufacturer’s instructions (Fontana-Masson Stain Kit, Abcam, Berlin, Germany). Samples were examined using either a Hamamatsu NanoZoomer S60 (Hamamatsu Photonics, Herrsching, Germany) or a bright field fluorescence microscope (Axio Imager.A1, Zeiss, Oberkochen, Germany) and images were processed using ProgRes CapturePro Software (JENOPTIK, Dresden, Germany).

### 3.3. Immunostaining of Frozen Sections

Immunostaining of frozen sections was performed as previously described [56]. Briefly, porcine tissue in OCT medium was cut into 6–8 µm sections, fixed in 4% paraformaldehyde (PFA) for 20 min, followed by permeabilization in 0.3% Triton X-100 in PBS for 10 min. The sections were blocked with 10% normal goat serum (NGS) or 10% normal donkey serum (NDS) and then incubated with primary antibodies (Appendix A) diluted in 1% NGS or NDS in PBS overnight at 4 °C or 2 h at room temperature. Alexa-fluor-488, -568, or -647-conjugated anti-mouse or -rabbit immunoglobulins (Life Technologies, Carlsbad, CA, USA) were used for detection, and nuclear staining was performed with 4′,6-diamidino-2-phenylindole (DAPI) (Vectashield antifade mounting medium with DAPI; Vector, Burlingame, CA, USA). Immunolabeled cryosections were examined with a laser scanning confocal microscope (TCS SP-8, Leica, Wetzlar, Germany). For negative controls, the primary antibodies were replaced by equimolar concentrations of an irrelevant isotypic primary antibody of the same species.

### 3.4. Immunostaining of Paraffin Sections

Immunohistochemistry was performed on paraffin sections as previously described [49]. The list of antibodies is provided in Appendix A.

### 3.5. Transmission Electron Microscopy

For TEM, tissue specimens were processed as described previously [57]. Briefly, the samples were fixed in 2.5% glutaraldehyde in 0.1 M phosphate buffer, dehydrated, and embedded in epoxy resin according to standard protocols. Ultrathin sections were stained with uranyl acetate–lead citrate and examined with an electron microscope (EM 906E; Carl Zeiss Microscopy, Oberkochen, Germany).

## Figures and Tables

**Figure 1 ijms-24-07543-f001:**
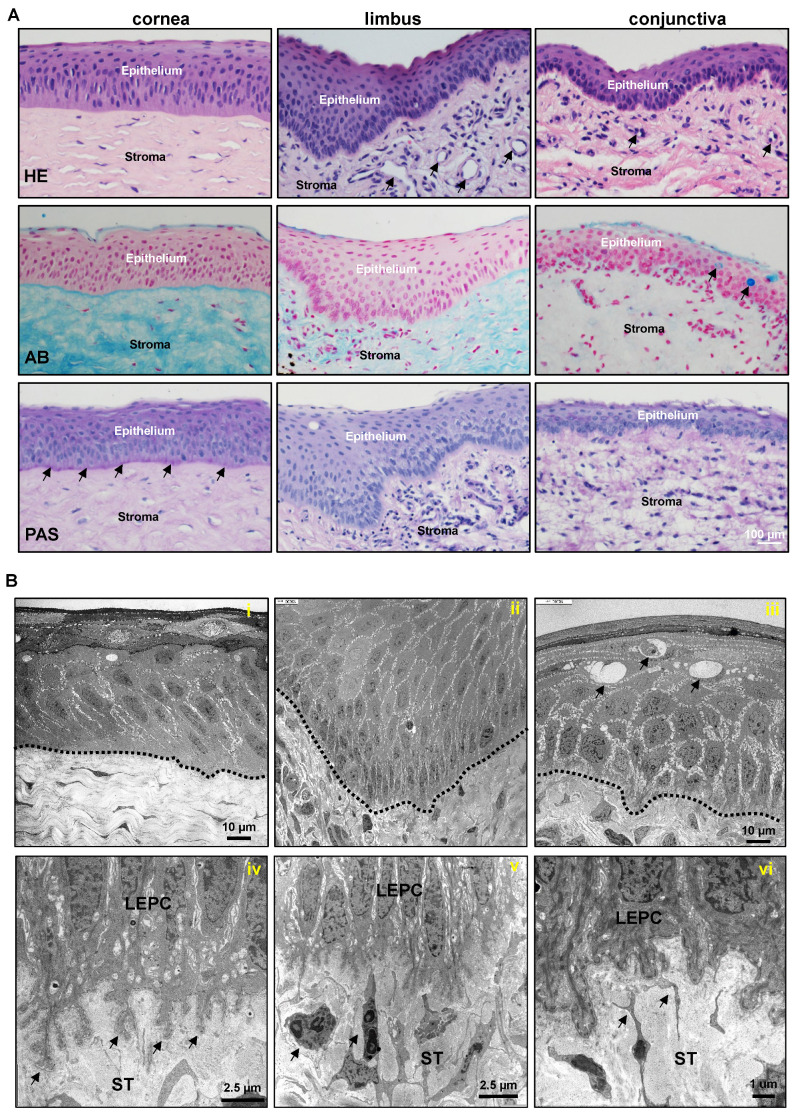
Histology of porcine ocular surface: (**A**) Light micrographs of paraffin sections of corneoconjunctival tissue stained with hematoxylin and eosin (HE) showing the corneal epithelium consists of 6–8 layers of stratified epithelium and Bowman’s layer is not evident; deep invaginations of the limbal epithelium (6–12 layers) analogous to the limbal interpalisade crypts of human limbal tissue can be observed in the limbal zone; the bulbar conjunctiva is composed of a 3–5 layered stratified epithelium; significant vessels (arrows) in the limbal and conjunctival stroma. The conjunctival and limbal stroma are organized differently to the corneal stroma, with loose connective tissue, a large number of fibroblasts, and the significant presence of vessels (arrows). Alcian blue (AB) staining of ocular surface showing the presence of glycosaminoglycans (blue) throughout the cornea and rather weak staining in the limbal and conjunctival stroma and the presence of goblet cells in the conjunctival epithelium (arrows). Periodic acid-Schiff (PAS) staining highlighting the BM of porcine corneal epithelium (arrows) and less pronounced in the epithelial BM of limbus and conjunctiva. (**B**) Transmission electron micrographs showing the architecture of the corneal (**i**), limbal (**ii**), and the presence of goblet cells in the conjunctival epithelia (arrows, (**iii**)) (dotted line represents basement membrane); extensive basal cell processes of limbal epithelial progenitor cells (LEPC) extending into the limbal stroma (ST) (arrows, (**iv**)), close interactions with stromal immune cells (arrows, (**v**)) and with mesenchymal stromal cells (arrows, (**vi**)).

**Figure 2 ijms-24-07543-f002:**
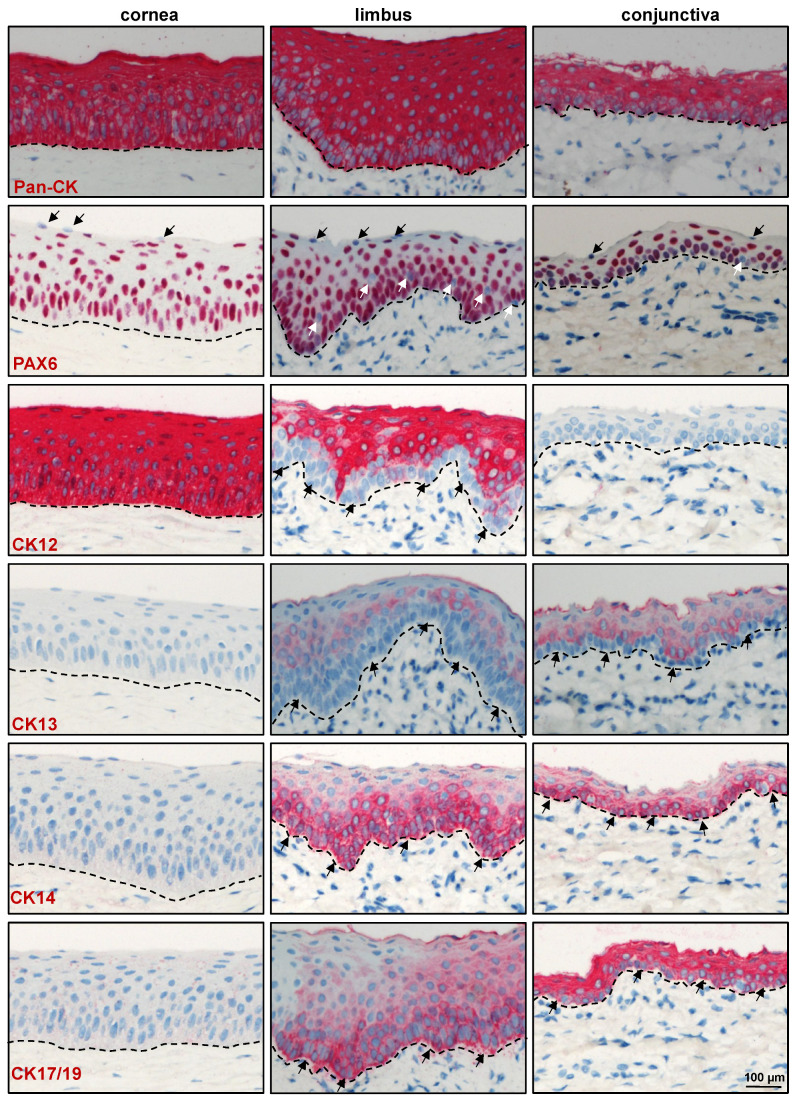
Immunohistochemical analysis of epithelial-related markers: Immunohistochemical staining of paraffin sections showing the expression of epithelial keratins (pan-CK) in all epithelial layers of the cornea, limbus, and conjunctiva (red); PAX6 expression in all layers of corneal epithelium and in nearly all epithelial cell layers of limbus and conjunctiva except for some basal cells (white arrows) and superficial epithelial cells (black arrows); cytokeratin (CK) 12 expression in all layers of the corneal epithelium, but only in suprabasal cell layers of limbal epithelium (arrows); CK13 expression in the suprabasal cell layers of both limbal and conjunctival epithelium and negative in the basal layers (arrows); CK14 expression in the basal layers of the limbal and conjunctival epithelia (arrows) but negative in the corneal epithelium; and CK17/19 expression in basal layers of the limbal epithelium (arrows) and in all layers of the conjunctival epithelium (arrows); dashed line represents basement membrane.

**Figure 3 ijms-24-07543-f003:**
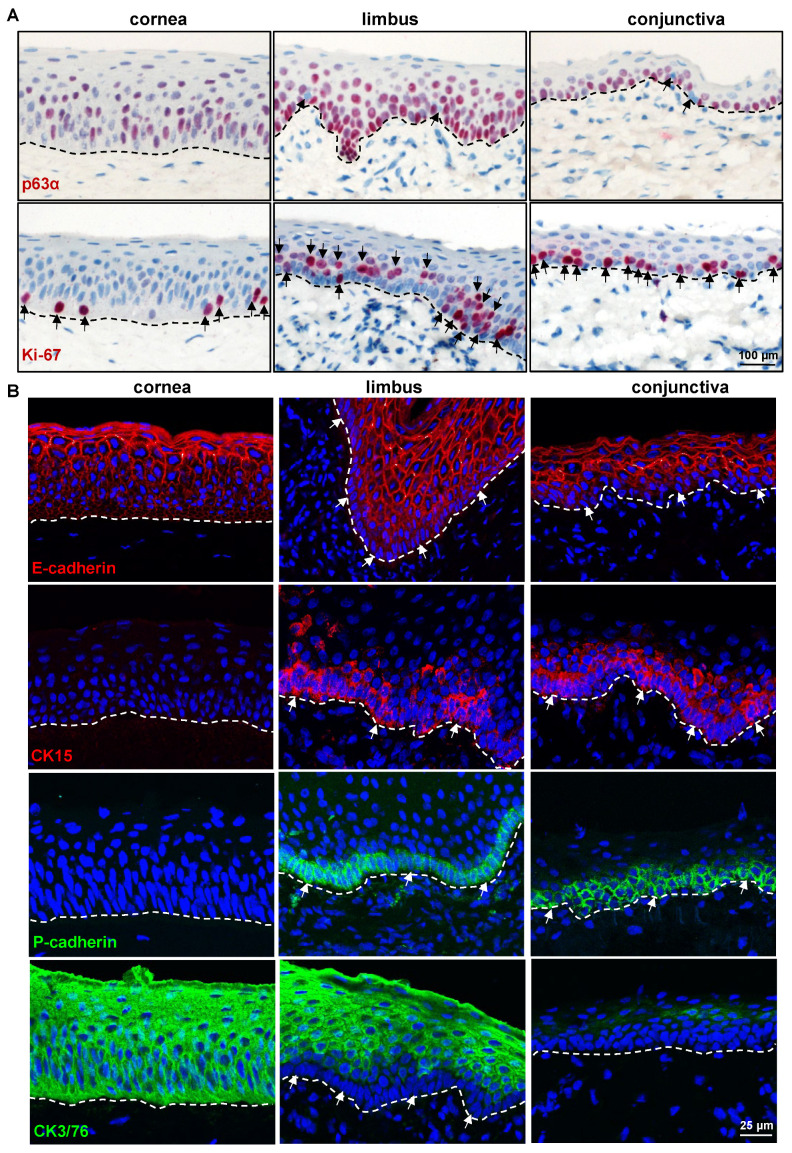
Expression analysis of epithelial-related markers: (**A**) Immunostaining of corneoconjunctival tissue showing the expression of p63α in the basal layers of corneal, limbal, and conjunctival epithelia, but the stronger expression in the limbus and conjunctiva and a few negative cells in the limbal and conjunctival basal layers (arrows); Ki-67 expression in the basal cells of the cornea, limbus, and conjunctiva (arrows); dashed line represents basement membrane (BM). (**B**) Immunofluorescence staining of frozen sections showing the expression of E-cadherin in the superficial layers of epithelial cells in both limbus and conjunctiva (red), leaving the basal cells negative (arrows), whereas all layers of corneal epithelium expressed E-cadherin; CK15 expression in the basal layers of the limbal and conjunctival epithelia (arrows, red) but not in the corneal epithelia; P-cadherin expression in the basal layers of limbal and conjunctival epithelia (arrows, green); CK3/76 expression (green) in all layers of the corneal epithelium and in suprabasal cell layers of limbal epithelium but not in basal layers (arrows); nuclear counterstaining with 4′,6-diamidino-2-phenylindole (blue); dashed line represents BM.

**Figure 4 ijms-24-07543-f004:**
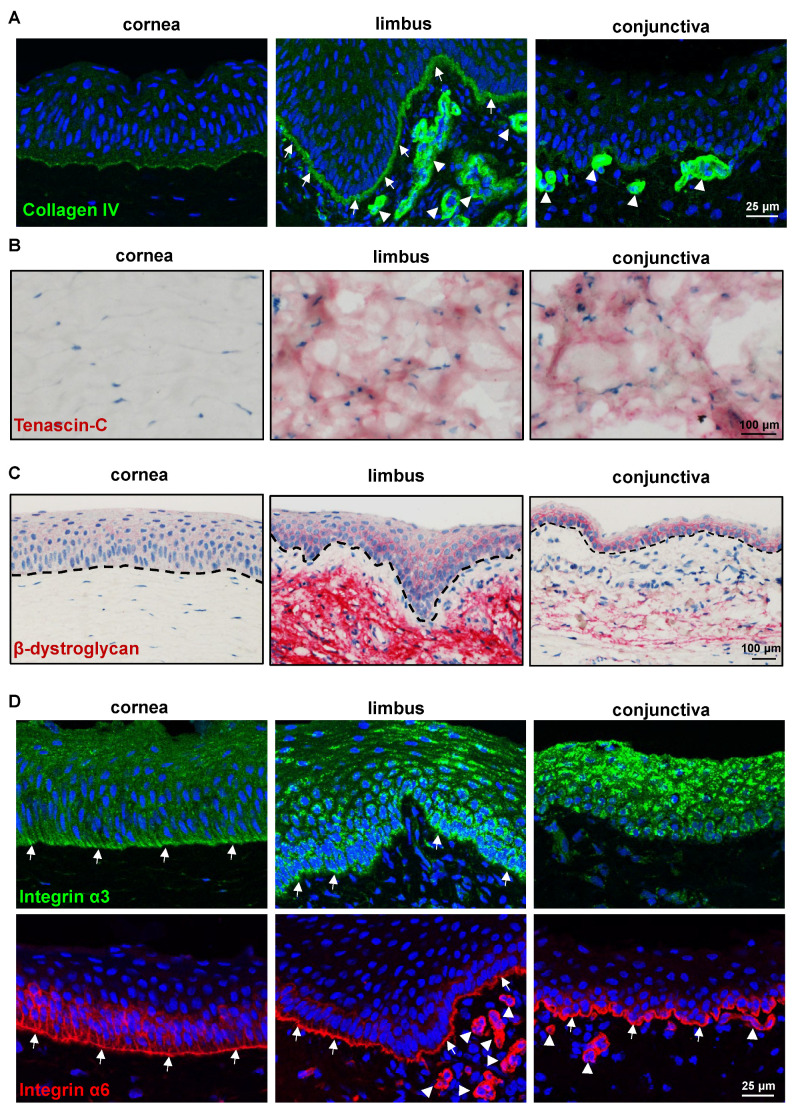
Expression analysis of extracellular matrix and adhesion molecules: (**A**) Immunohistochemistry and laser confocal imaging of corneoconjunctival tissue sections showing the collagen IV strongly stained limbal epithelial basement membrane (BM) (arrows); very weak staining in corneal and conjunctival epithelial BM; strong staining in the limbal and conjunctival vessel BM (arrowheads). Nuclei are counterstained with 4′,6-diamidino-2-phenylindole (DAPI, blue). (**B**) Immunohistochemical analysis of corneoconjunctival tissue showing the tenascin(TN)-C expression in the stroma of limbus and conjunctiva (red) but not in the corneal stroma. (**C**) Immunohistochemical staining of corneoconjunctival tissue sections showing the β-dystroglycan expression in the basal cell membranes of the limbus and conjunctiva; faint membrane staining in the superficial cells of the limbus and conjunctiva as well as corneal epithelium; prominent staining in the limbal stroma but weak in the conjunctival stroma and absent in the corneal stroma. The dashed line represents BM. (**D**) Immunohistochemistry and laser confocal imaging of corneoconjunctival tissue sections showing the integrin α3 expression in the basal aspect of corneal, limbal, and conjunctival epithelium, with the limbus and conjunctiva showing a strong expression (green); integrin α6 expression in all ocular surface epithelia, with a continuous strong basal membrane staining (arrows) and prominent staining in the vasculature of limbal and conjunctival stroma (arrowheads); nuclei are counterstained with DAPI (blue).

**Figure 5 ijms-24-07543-f005:**
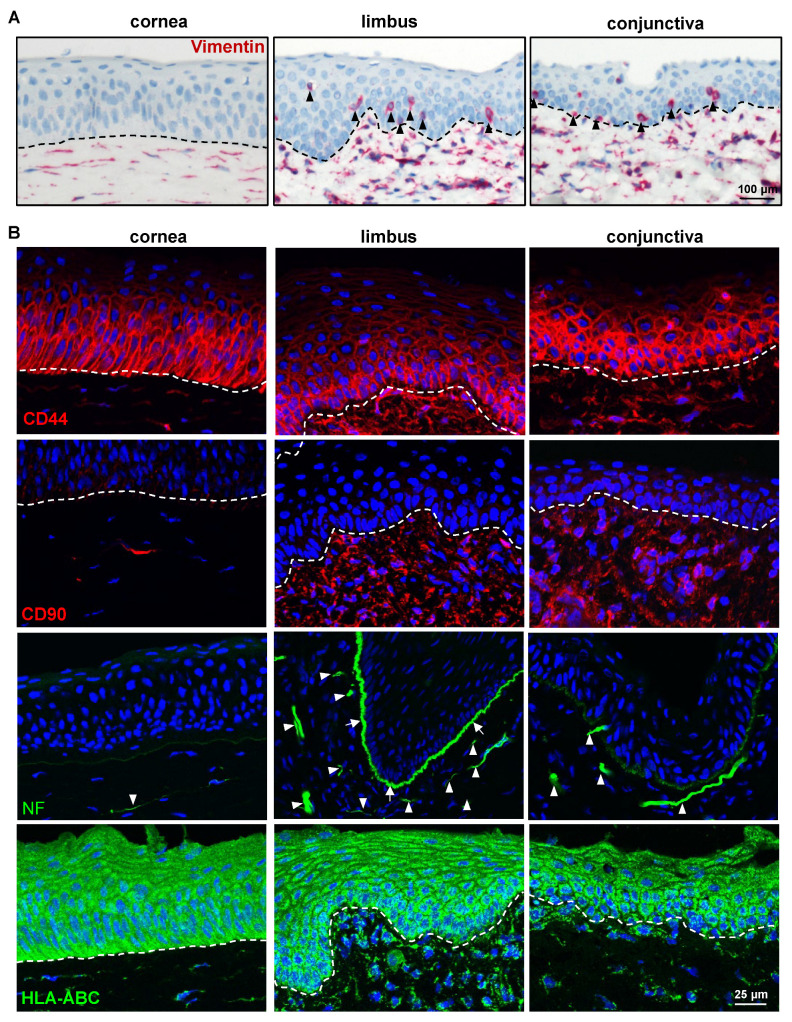
Expression analysis of limbal niche cell-related markers: (**A**) Immunohistochemical staining of paraffin sections of corneoconjunctival specimen showing the vimentin expression in the stromal cells of the porcine cornea, limbus, and conjunctiva (red); vimentin^+^ cells in the basal epithelial layers of both limbus and conjunctiva (arrowheads); dashed line represents the basement membrane (BM). (**B**) Immunofluorescence staining corneoconjunctival tissue sections showing the CD44 expression on the plasma membranes of basal epithelial and stromal cells of the cornea (red) and predominant expression in the limbus and conjunctiva; CD90 expression in the stromal cells of limbus and conjunctiva (red); neurofilament (NF) expression in the stroma of all three surface regions (arrowheads) and in the limbal BM (arrowheads); the anti-human HLA-class I antibody stained all cells of the ocular surface (dashed line represents BM).

**Figure 6 ijms-24-07543-f006:**
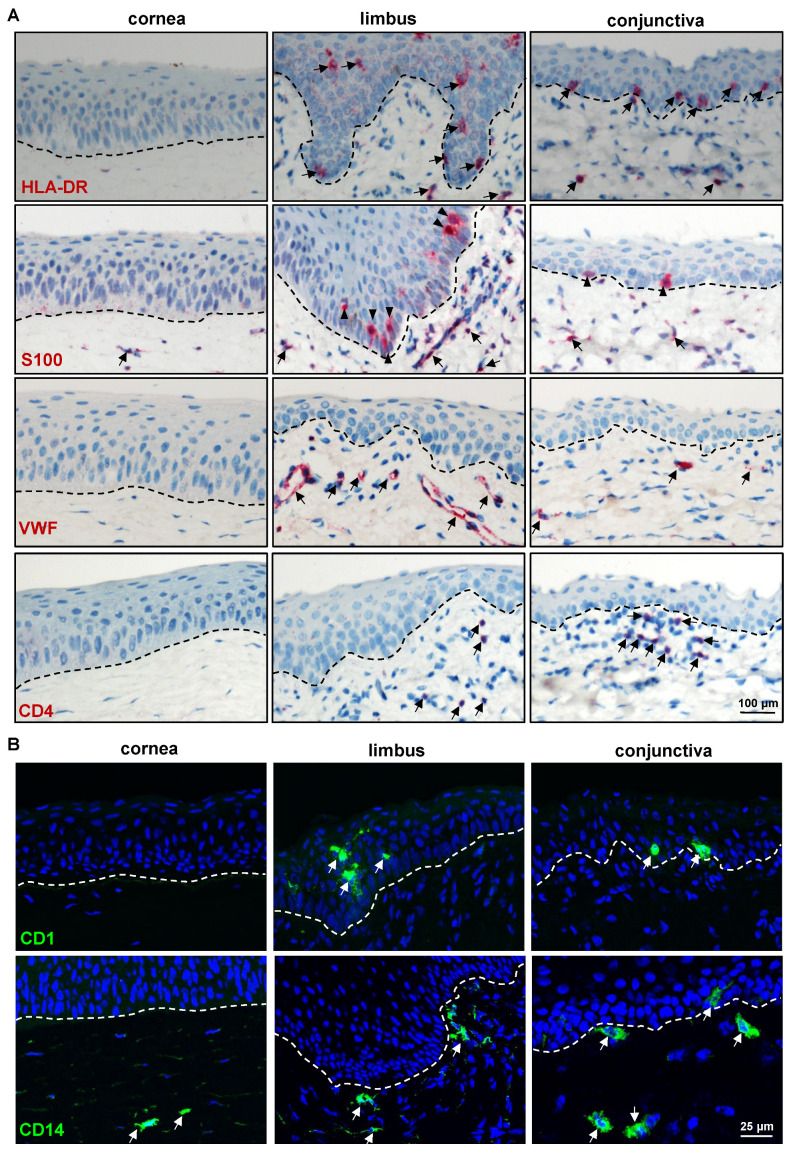
Expression analysis of immune-related markers: (**A**) Immunohistochemical staining of porcine ocular surface paraffin tissue sections showing the swine leukocyte class II^+^ cells (detected by HLA-class II antibody) in the epithelial layers and stroma of limbus and conjunctiva (arrows); S100 expression in a few cells of the limbus and the conjunctival epithelial basal layer (arrowheads) and in the stroma of all three regions (arrows); anti-von Willebrand factor (VWF) stained the vasculature of the porcine ocular limbus and conjunctiva (arrows); expression CD4 in a few cells of limbal and conjunctival stroma; dashed line represent basement membrane (BM). (**B**) Immunofluorescence staining of frozen sections showing the CD1^+^ cells within the multilayered epithelium of limbus and conjunctiva (arrows) but not in the cornea; CD14 expression in a few cells of the limbal and conjunctival stroma (arrows), rarely observed only in the conjunctival epithelium (arrow); nuclei are counterstained with 4′,6-diamidino-2-phenylindole (blue); dashed line represents the BM.

**Figure 7 ijms-24-07543-f007:**
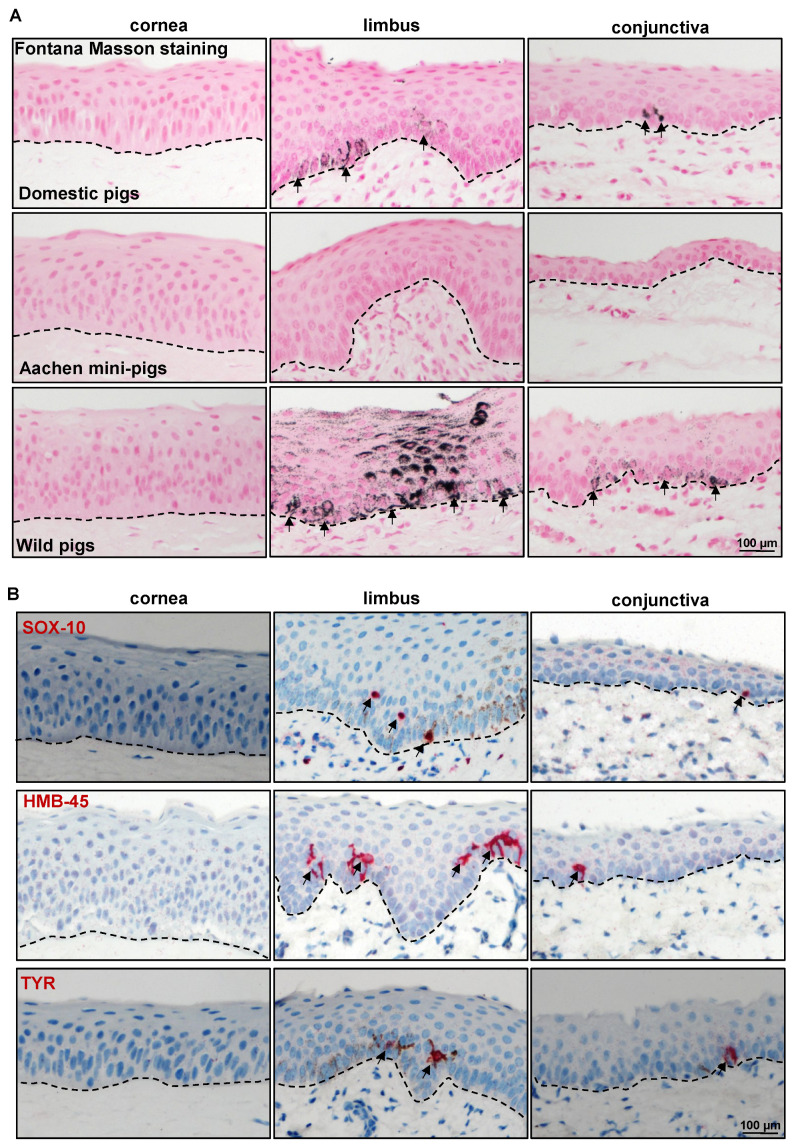
Expression analysis of melanocyte-related markers: (**A**) Light micrographs of paraffin sections of corneoconjunctival specimen stained with Fontana-Masson showing the melanin in the limbal and conjunctival regions of domestic porcine samples and wild pigs (arrows), whereas there was no melanin in the Aachen minipig samples; dashed line represents the basement membrane (BM). (**B**) Immunohistochemical staining corneoconjunctival sections showing the SRY-homeobox10(SOX-10)^+^, human melanoma black(HMB-45)^+^, and tyrosinase (TYR)^+^ cells in the basal layers of limbal and conjunctival epithelium; dashed line represents BM.

**Figure 8 ijms-24-07543-f008:**
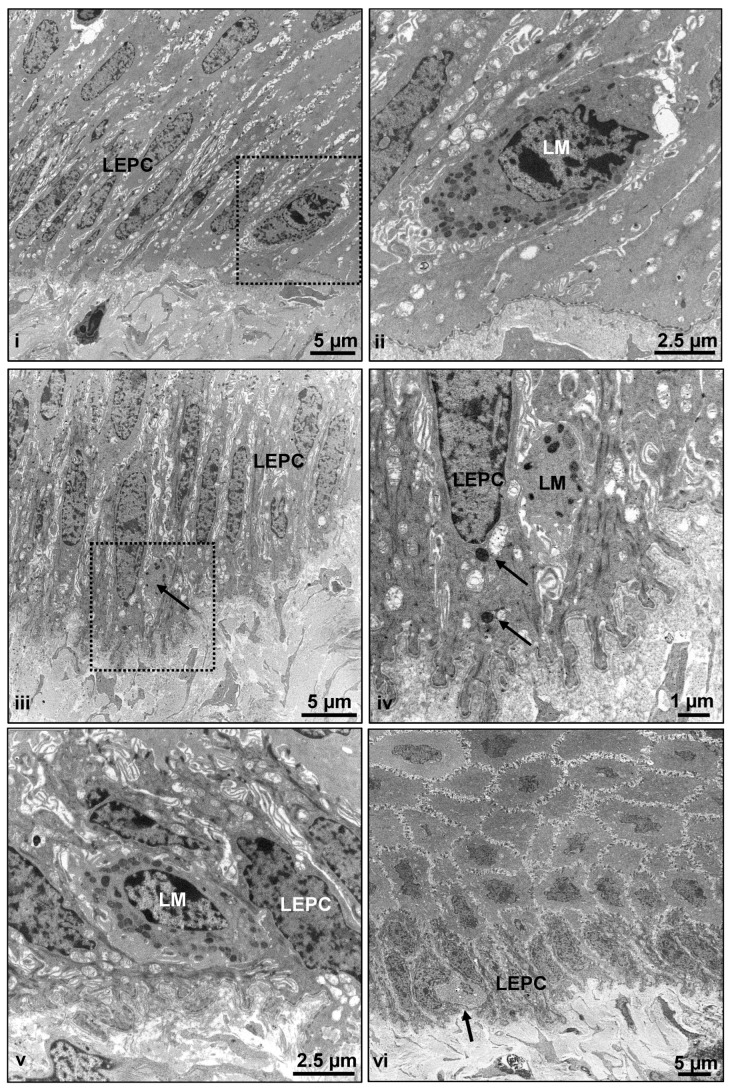
Electron microscopy analysis of limbal melanocytes: Transmission electron micrographs showing basal limbal epithelial cells (LEPC) in association with limbal melanocytes (LM) at the corneo-limbal transition zone (**i** (recatangle), **ii**) and limbal palisade zone (**iii** (arrow in the recatangle showing the melanocyte)–**v**); melanosomes within LEPC are shown in (**iv**) (arrows); atypical cells in the basal layer of limbal epithelium assuming atypical melanocytes (arrow, (**vi**)).

**Table 1 ijms-24-07543-t001:** A list of epithelial markers and their expression patterns on porcine ocular surface epithelial cells.

	Cornea		Limbus		Conjunctiva	
	Basal	Suprabasal	Basal	Suprabasal	Basal	Suprabasal
CK3	++	++	−	++	−	−
CK12	++	++	−	++	−	−
CK13	−	−	−	(+) or +	−	++
CK14	−	−	++	-	++	−
CK15	−	−	++	-	++	−
CK17/19	−	−	++	-	++	+
E-cadherin	+	++	− or +	++	− or +	++
Epithelial keratins (Pan-CK)	++	++	++	++	++	++
Integrin α3	+	(+)	+	(+)	+	+
Integrin α6	++	−	++	−	++	−
p63α	(+)	−	++	−	++	−
Pax6	++	++	++	++	++	++
P-cadherin	−	−	+	−	+	−

−, undetectable; (+), weak positivity; +, moderate positivity; ++, strong positivity.

## Data Availability

The datasets generated during and/or analyzed during the current study are available from the corresponding author on reasonable request.

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
