# Peer review of "Characterization of Porcine Ocular Surface Epithelial Microenvironment"

_ijms, 2023, doi:10.3390/ijms24087543_

Round 1
Reviewer 1 Report
Intro
A comprehensive histological overview of the ocular surface epithelium in the domestic pig. Requires more critical analysis of the results. Nice images presented neatly.
General: a few typos, a few areas where an acronym is used (write out in full the first time)
Intro:
Needs more synthesis of the current literature (not necessarily new references)- what is known about porcine ocular surface (OS) epithelium, what was unknown before this study.
Ocular surface (pig cornea) is used to model multiple diseases and test for drug permeation etc. It seems limiting to just focus on LSCD and suggest opening this up as it knowledge of the OS epithelial microenvironment is relevant to other diseases (and a broader authorship).
Explain what specifically you mean by OS epithelial microenvironment.
Results:
General: dashed line is not on all images- include it.
In some cases, it is very difficult to determine where the superficial layer is- may be worth including a second line in these images to mark the actual surface (e.g fig 3B)
Provide a summary at the start of each sub section.
For all the images: mention somewhere in the results that these are representative images and the number of sections/animals which were examined (5 I think) for each marker (suggest in section 2.2).
Section 2.1
Fig 1A (suppl) cf to Crespo-Moral ref?
Fig 1A (in paper) ref the arrows as blood vessels and also as goblet cells- correct this.
Section 2.2
Supply a better overview of the results, summarise (aim 41 markers, but X did not work, so used Y).
Section 2.3
Use Pax6, not PAX6.
CK12: provide better description of differential staining seen between the limbus and the conjunctiva
CK13 (Suppl fig 2)- over expressed- suggest a lower exposure/ rationalise why overexposed.
CK17/19 – in the limbus image can’t see the superficial surface- provide an image which contains it.
P63-alpha- also mentioned p63- make sure same.
Ki67 – mention seen in a monolayer in cornea and conjunctiva vs in multiple layers in limbus. Instead of “more present – line 179- suggest “more frequently detected”
Fig 2: some of the conjunctiva samples look a little different (thicker vs thinner samples) comment on this somewhere.
E-cadherin- not convinced that the basal cells are negative in the limbus.
Table 1:
Why is E-cadherin here twice? Suggest including a final column with comment cf human OS epithelium -e,g, same or different cell types (qualitative, not quantitative)
Section 2.4 and 2.5:
Include tables 2 and 3 similar to table 1. Within this, include a final column, stating what differs compared to the human
Section 2.4:
Need a sentence summarising the findings here.
Tenascin- submit similar image to the rest, including epithelial layers- this looks like a stroma only image and (while it might not be) it looks like it is higher magnification- confirm.
Section 2.5:
Need to mark the superficial epithelial layer on images in fig B (where it is not obvious
CD44: Comment on low expression in corneal stroma.
NF staining- idea about cross-reactive binding or artifact- need to come back to this in the conclusion- was this present in all animals examined?
HLA-Abc – stained all epithelial cells,? Not all of OS.
CD4: cell highlighted are purple, not red (would appear by the “CD4” written on image in red, that red was anticipated?
Melanocytes: more on differences between mini pigs, wild pigs and domestic pigs- are they very different genetically?
Line 380: ref arrows in fig 7B- is this meant to ref 7A? I thought 7B was domestic pigs, not wild pigs.
Fig 8- is there meant to be an arrow in (iii)?
Conclusion: requires a more in depth contextualisation of this study to the current literature.
What new things are now known about pig OS epithelium? What differences exist between pigs and humans and give an overview of the limitations of the current study. Mention other methods which could be used to cf porcine and human OS epithelium (e.g. transcriptomics/proteomics methods)
Proposed future directions- e.g. cf mini pigs and wild pigs.
Refs:
fine
Methods:
Ethics statement required (or exemption from)
There is still a risk that not detected does not mean not expressed (the experiment could have failed technically). Positive controls (human samples) could potentially have been run- perhaps address this during the discussion (e.g. non availability of these samples).
Reviewer 2 Report
In the current study, authors have characterized porcine ocular surface through immunohistological methods. Authors have done so by validating human targeted antibodies to study porcine ocular surface and suggested their potential use in investigating ocular diseases. So far, this is a very thorough study that analyzed the differential localization of 41 protein markers at various ocular substructures in the porcine model and significantly contributes to the existing literature in the field of ophthalmology. Overall study is well designed, and the manuscript is well written with few grammatical errors. Minor corrections are:
Scale bar is missing for few of the images in the Fig 1A.
What is the basis of a separate results section, 2.2? 2.2 can be merged into 2.3.
